# New Devices, Innovative Technologies, and Non-Standard Techniques for Airway Management: A Narrative Review

**DOI:** 10.3390/healthcare11182468

**Published:** 2023-09-05

**Authors:** Tomasz Gaszyński, Manuel Ángel Gómez-Ríos, Alfredo Serrano-Moraza, José Alfonso Sastre, Teresa López, Paweł Ratajczyk

**Affiliations:** 1Department of Anesthesiology and Intensive Therapy, Medical University of Lodz, 90-154 Lodz, Poland; pawel.ratajczyk@umed.lodz.pl; 2Department of Anesthesiology and Perioperative Medicine, Complejo Hospitalario Universitario de A Coruña, 15006 A Coruña, Spain; magoris@hotmail.com; 3Emergency Medical Service EMS SUMMA 112, 28045 Madrid, Spain; alfchus@gmail.com; 4Complejo Asistencial Universitario de Salamanca, 37001 Salamanca, Spain; josealfsastre@hotmail.com (J.A.S.); teresina1234@hotmail.es (T.L.)

**Keywords:** airway management, difficult airway, endotracheal intubation, clinical practice

## Abstract

A wide range of airway devices and techniques have been created to enhance the safety of airway management. However, airway management remains a challenge. All techniques are susceptible to failure. Therefore, it is necessary to have and know the greatest number of alternatives to treat even the most challenging airway successfully. The aim of this narrative review is to describe some new devices, such as video laryngeal masks, articulated stylets, and non-standard techniques, for laryngeal mask insertion and endotracheal intubation that are not applied in daily practice, but that could be highly effective in overcoming a difficulty related to airway management. Artificial intelligence and 3D technology for airway management are also discussed.

## 1. Introduction

Airway management is as old as medicine itself. The first references to airway treatment come from ancient Egypt, more than 3500 years before Christ, with the description of a tracheostomy [1]. Since then, the discipline has undergone a profound evolution, making available multiple devices and a considerable amount of knowledge based on the research and derived scientific evidence. The development of airway management techniques and devices has been influenced by the progress of medical knowledge and technological advancements. A wide range of airway devices and techniques have been created over the past century to enhance the safety of airway management. Thus, there are various advancements, beginning with the advent of the laryngoscopy in the 1940s, followed by fiberoptic laryngoscopy in the 1960s, the introduction of supraglottic airway devices in the 1980s, and, lastly, the emergence of modern video laryngoscopy in the 2000s [2]. However, airway management remains a challenge. All techniques are susceptible to failure. Therefore, it is necessary to have and know the greatest number of alternatives to treat even the most challenging airway successfully [3].

The achieved advancements demonstrate their potential in enhancing success rates and minimizing the adverse events associated with airway instrumentalization. Establishing an individualized and structured airway management plan for each patient is essential. Technology can assist in both evaluating the airway and facilitating tracheal intubation [4]. Likewise, the combination of multiple airway devices, also known as a multimodal approach [5], is a successful practice used in complex airways. However, these techniques are often underused.

We conducted a narrative review due to the insufficient scientific evidence available for each technique, device, and novel technology included, making it impractical to perform a structured systematic review. To accomplish this, we conducted a search in the MEDLINE/PubMed databases utilizing the keywords “difficult airway” and “techniques”, “difficult airway” and “devices”, and “difficult airway” and “technologies” within the timeframe of 1 January 2010 to 30 April 2023. The screening of these records was conducted based on their titles or abstracts. Exclusions were performed for publications not in the English language and those involving non-human studies. All articles with the potential for relevance were procured in their entirety and subjected to comprehensive evaluations by TG, MGR, AS, and TL. Additionally, the reference sections of each article were meticulously examined to identify other potentially pertinent studies, encompassing randomized controlled trials, prospective studies, and retrospective studies. Any disparities were settled through consensus among the reviewers. For those specific devices/techniques where the evidence is lacking, the authors provide their own first-hand experiences.

The aim of this article is to describe some new devices and techniques that are not applied in daily practice, but that can be highly effective in overcoming a difficulty related to airway management.

## 2. New Devices

In recent times, the introduction of two new devices stands out in the market: video laryngeal masks and articulated stylets.

### 2.1. Video Laryngeal Masks

Video laryngeal masks (VLMs) represent an advancement in second-generation supraglottic airway devices (SADs). These devices involve the integration of a videoscope into a second-generation laryngeal mask, enabling the visualization of the previously unobservable process of SAD insertion and placement. Thus, VLMs are a comprehensive and versatile supraglottic airway device that comprise two components [6]: (1) a disposable second-generation SAD, which features an anatomically curved shape, separate gastric and ventilation channels for functional separation, a built-in bite block, a silicone cuff, and a reinforced distal tip that enables a tighter seal and higher oropharyngeal sealing pressure; and (2) a reusable flexible videoscope equipped with a built-in or stand-alone display. The videoscope is inserted into a closed-tip channel within the SAD, specifically designed to securely accommodate it, thus ensuring the reusable part remains isolated from direct patient contact to prevent contamination. Although VLMs share many characteristics with the latest SADs, their inclusion of this enhancement has earned them the classification of “third-generation SADs” [6]. This configuration allows the device to incorporate several desirable characteristics: the possibility of vision-guided insertion, corrective maneuvers when needed, precise placement in the hypopharynx, gastric tube insertion, and endotracheal intubation under visual guidance. Therefore, VLMs combine the advantages of an intubating laryngeal mask and a video laryngoscope. These properties enable the establishment of effective ventilation and optimal oxygenation throughout the tracheal intubation process, resulting in reduced apnea duration. The inflation of the cuff creates a secure seal, safeguarding the airway against gastric aspiration and maintaining its clearance from blood or secretions until tracheal intubation is accomplished. The enhanced view of the glottis significantly improves the success rate of first-attempt tracheal intubation in comparison to a blind technique, which is never desirable.

The SaCo Video Laryngeal Mask (SaCoVLM™, UE MedicalR, Zhejiang, China) [7], the Totaltrack VLM (MedComflow S.A., Barcelona, Spain) [8], and the LMA CTrach (The Laryngeal Mask Company, Le Rocher, Victoria, Mahe, Seychelles) [9] are the main VLMs that have been introduced into the market and extensively studied in the clinical setting.

### 2.2. Articulated Stylets

Stylets are essential classic adjuncts for tracheal intubation as they allow preconfiguring the shape of the endotracheal tube and thus guiding it towards the glottis [10]. However, when there is an acute angle between the distal tip of the endotracheal tube and the glottis, these devices are unable to overcome the difficulty in securing the airway. This problem is particularly common with video laryngoscopy since, although it improves the indirect visualization of the glottis compared to direct laryngoscopy, there is often a difficulty in directing the endotracheal tube and inserting it through the vocal cords. The articulated stylets were introduced into the market to overcome this drawback. These devices are characterized by their ability to articulate their distal ends from their proximal ends, which allows negotiating the angle between the tip of the endotracheal tube and the glottis. Potentially, these adjuvants can increase the rate of tracheal intubation on the first attempt and instrumentation in the airway [11].

The Flextip (P3Medical Ltd., Bristol, UK) [12], the Parker Flex-It Directional Stylet (Parker, Highlands Ranch, CO, USA) [11], the Total Control Introducer (TCI; Through The Cords, LLC, Salt Lake City, UT, USA) [13], D-FLECT^®^ deflectable-tip bougie (Specialist Airway Solutions Pty Ltd., Brisbane, Australia) [14], the Truflex™ articulating stylet (Truphatek International Limited, Netanya, Israel) [15], the ProVu^TM^ articulating Video Stylet, (Flexicare Inc., Irvine, CA, USA) [16], and the StyletScope™ fibreoptic stylet (FOS; Nihon Koden Corporation, Tokyo, Japan) [17] are examples of this type of device.

Another interesting approach for addressing the challenges associated with the passage of a tube through the vocal cords during laryngoscopy is the combined use of a laryngoscope and a bronchoscope, known as combined laryngo–bronchoscope intubation (CLBI) [18,19]. This technique is also commonly referred to as the “smart stylet” technique [20]. In this method, the bronchoscope is employed to articulate the guide for the endotracheal tube—not for visualizing the glottis. Meanwhile, the laryngoscope or video laryngoscope is utilized to visualize the entrance to the larynx and to direct the bronchoscope toward the glottis (Figure 1).

This technique is referred to as video-assisted endoscopic intubation (VAEI) and can be performed with a wide range of video laryngoscopes [18,19,20,21,22,23,24].

## 3. Non-Standard Techniques

### 3.1. Laryngeal Mask Insertion

#### 3.1.1. Bougie-Assisted Laryngeal Mask Insertion

The ProSeal LMA (pLMA, Teleflex Incorporated, Wayne, PA, USA) tube is soft and can bend during insertion, making placement difficult. The insertion of this supraglottic airway guided by a gum elastic bougie is associated with a higher success rate compared to the standard digital technique and the introducer technique; thus, this method is well described in the literature [25,26,27]. Second-generation LMAs are anatomically shaped and curved and the ventilation channel material is more robust. These features facilitate insertions with classical techniques. However, they may also benefit from a bougie-assisted or catheter-assisted technique when the insertion is difficult, solving the problem. Therefore, it is worth remembering this technique. There are at least two possible bougie-assisted laryngeal mask insertion techniques. The first one consists of introducing the distal end of the bougie into the esophagus blindly or with the use of a laryngoscope. Subsequently, its proximal tip is inserted into the distal tip of the laryngeal mask suction channel. The laryngeal mask is then inserted into the pharynx following the bougie as a guide wire. The second technique involves placing the bougie in the suction channel of the laryngeal mask so that most of the length of the bougie is outside the distal tip of the laryngeal mask before inserting it into the patient’s mouth. Subsequently, the distal tip of the bougie is inserted into the esophagus to “railroad” the laryngeal mask onto the introducer (Figure 2). The authors used a gastric catheter-assisted insertion of a laryngeal mask (Figure 3). The gastric catheter is inserted into the gastric channel of the laryngeal mask prior to mask placement. The gastric catheter is inserted in the esophagus of the patient and the laryngeal mask follows the catheter as a “guide wire”. The additional advantage of this technique may also be the immediate evacuation of air from gastric insufflation after the insertion of the laryngeal mask.

#### 3.1.2. Spoon-Assisted Laryngeal Mask Insertion

Narrow oral cavities may result in a difficult laryngeal mask insertion. This inconvenience is caused because, when inserting the device into the oral cavity, the tongue is placed on the laryngeal mask blocking its way. Likewise, the tip of the laryngeal mask can be bent on its axis. Spoon-assisted laryngeal mask insertion, although it sounds strange, was found to be very effective to avoid this problem. The spoon is inserted into the mouth of the patient to elevate the tongue and create more space to insert the supraglottic airway.

#### 3.1.3. Assisted Laryngeal Mask Insertion with Other Airway Devices

“Blind” insertion of supraglottic airway devices results in malposition in the pharynx in 50–80% of the cases [28]. Consequently, this can lead to suboptimal airway control due to inadequate oropharyngeal sealing, obstruction or air leaks, increased risk of displacement, poor gas exchange, airway trauma, and morbidity [29,30]. Therefore, achieving the correct location is a priority [6]. Van Zundert et al. [31] demonstrated that the vision-guided insertion technique using an “insert–detect–correct-as-you-go” technique under the vision of a video laryngoscope and standardized corrective measures could achieve almost a 100% correct positioning of a supraglottic airway. Therefore, in the absence of new video laryngeal masks [6], a multimodal approach may be the solution.

The laryngoscope-guided insertion of laryngeal masks showed an improved seal pressure of the laryngeal mask; although, it increased the complexity of the procedure and prolongs the insertion time [32].

In a similar fashion, the combined use of video laryngoscopes optimizes the positioning of the laryngeal mask. This confers an increase in the sealing pressure and a decrease in complications [33,34]. The technique of the video laryngoscopy-assisted insertion of the laryngeal mask is similar to video laryngoscopy tracheal intubation. The video laryngoscope is inserted into the mouth to visualize the entrance to the larynx (Figure 4A). The laryngeal mask is then inserted and the placement of the laryngeal mask cuff is observed in the monitor (Figure 4B).

Likewise, the combined use of visual stylets and video-intubation stylets has been described for this purpose [35]. From our experience, the use ProVu™ Video Stylet (Flexicare Medical Ltd., Mountain Ash, UK) constitutes an interesting method. The video intubation stylet is placed in the ventilation channel of the laryngeal mask so that its optical tip is located at the distal end of the ventilation channel. When the laryngeal mask is inserted, its position in the pharynx can be observed on the video stylet monitor. Finally, the stylet is removed after the correct placement of the supraglottic airway device.

### 3.2. Endotracheal Intubation

#### 3.2.1. Tracheal Intubation with Video Laryngoscopy and Nonstandard Aids

Video laryngoscopy plays a fundamental role in airway management since its incorporation into clinical practice. This technology is highly effective since it allows achieving a good glottic view on the first attempt in 98% of the cases [36]. However, visualization does not guarantee successful tracheal intubation [37]. This is one of its main disadvantages. The difficult passage of the endotracheal tube through the vocal cords, despite a better visualization of the glottis compared with direct laryngoscopy, has been extensively described. Conventional maneuvers, such as the use of stylets or bougies, can be useful to overcome this problem. Nonetheless, the advancement of the endotracheal tube may be impossible, since these adjuvants do not allow the manipulation of the distal tip of the endotracheal tube and may strike the anterior tracheal wall due to the acute stylet’s angle. A multimodal approach with other airway devices as fiberscopes or video stylets allows solving this common problem of indirect laryngoscopy. However, these techniques are still underutilized in the clinical context.

The use of a video laryngoscope plus a fiberscope provides several advantages. Both types of devices may compensate each other’ limitations in patients, such as those with a large tongue, obesity, or a cervical mass [38]. Thus, the video laryngoscope provides an unobstructed airway for the fiberscope, placing its tip in the vicinity of the glottis, and provides a visualization of the advancement of the endotracheal tube over the fiberscope, while the fiberscope negotiates the sharp angle between the endotracheal tube tip and the glottis [24].

The combined use of a video laryngoscope and a video intubation stylet is called the “video-twin technique” [39]. If the video laryngoscope does not provide adequate an visualization of the glottis, it is used to visualize the epiglottis and the insertion of the video stylet below the epiglottis under direct visualization. This technique provides better visualization and requires less force to elevate the tongue and epiglottis, less space to introduce the video laryngoscope blade into the mouth, and can result in fewer airway complications.

The Goldfinger (Ethicon Endo Surgery, Johnson & Johnson, Cincinnati, OH, USA) is a device designed for laparoscopic surgery. It is a stylet used for the right placement of gastric bands in bariatric surgery. However, it is widely used in several kinds of operations due to its properties. The Goldfinger has a flexible tip, which can be controlled by pushing a button on the handle. The device consists of a handle, a metal stem, and a tip. The tip can be angled up to 90 degrees on a vertical axis by pushing the button. Its rigidity depends on the position of the tip. When it is fully angled up to 90 degrees, the tip is rigid. Then, after the relaxation of the angle, it becomes flexible again. When tracheal intubations are attempted with a video laryngoscope, the endotracheal tube may need to be mounted on a short, typical J-shaped stylet. Sometimes the attempts to adjust the curvature of the guide are unsuccessful. Due to the sharp insertion angle of the endotracheal tube tip, resistance caused by striking against the anterior wall of the trachea can be experienced. The insertion of the Goldfinger into the endotracheal tube can enable the manipulation of its distal tip from the neutral position to 90 degrees (Figure 5A). While observing the larynx entrance with the video laryngoscope, the operator can flex the distal tip of the endotracheal tube to about 70 degrees by pushing the handle button and introducing the tube into the larynx inlet (Figure 5B).

Zeidan et al. [40] reported the utility of the Goldfinger for tracheal intubation with a Glidescope video laryngoscope in a morbidly obese patient with a difficult airway. The authors pointed out that the downward relaxation of the tip of the Goldfinger enabled the anesthetist to insert the tube further and avoid hooking onto the anterior wall of the trachea. Moreover, the risk of damage to the trachea was minimized.

These trials and clinical practice revealed that the availability of a maneuverable stylet are remarkably helpful during tracheal intubations with a video laryngoscope, where, despite good glottic view, it is not possible to advance the tube through the vocal cords.

#### 3.2.2. Endotracheal Intubation Using Custom-Made Video Laryngoscopes

Video laryngoscopes have an important role in airway management as a primary and rescue device. The COVID-19 pandemic has facilitated the universalization of the use of these devices [41]; although, there are hospitals where video laryngoscopes are not yet available. Despite the lower price due to the wide variety of devices on the market, video laryngoscopes will not be available in all ambulances. An interesting solution to this problem is creating a custom-made video laryngoscope. There are several papers on the use of custom-made video laryngoscopes in both manikin studies and clinical practice [42,43,44]. The advantage of a custom-made video laryngoscope is its price. A simple USB waterproof camera for smartphones costs no more than USD 20. The disadvantage is that these are no medical devices; therefore, they have no official approval to be used on patients. We tested these custom-made devices on a manikin model using a USB waterproof camera for smartphones attached to a standard single-use laryngoscope blade (Figure 6) (authors own unpublished data). Making a laryngoscope handle by using a 3D printer is another possibility. The necessary software is available on the Internet. This custom-made video laryngoscope works correctly in case of a difficult airway and the lack of availability of video laryngoscopes.

#### 3.2.3. Elective Tracheal Intubation for a Patient in the Lateral Position Using Video Airway Devices

In those patients who required the lateral decubitus position as the surgical position, anesthetic induction and tracheal intubation were previously performed in the supine position. However, the wide availability of video laryngoscopes allows for a general anesthesia induction and tracheal intubation of the so-called self-positioning of the patient in non-standard positions to eliminate the need for mobilization. There are several papers confirming the efficacy and safety of elective intubation in the lateral position [45,46,47,48]. This approach reduces the potential risk of complications related to the patient’s position, such as pressure on the nerves and vessels, or the dislocation of joints. The authors of this paper performed many successful tracheal intubations in patients anesthetized in the lateral position [48]. We used channeled and non-channeled video laryngoscopes for this purpose (Figure 7). All of them showed their usefulness in these settings. New third-generation supraglottic devices with a video camera as a Video Laryngeal Mask SaCo (UE Medical Corporation, Xianju, China) were a good alternative (Figure 8). This approach has an added advantage: tracheal intubation is performed after securing the airway and establishing optimal ventilation through the supraglottic airway.

#### 3.2.4. Elective Tracheal Intubation Using the Face-to-Face Technique

The face-to-face technique, also known as inverse intubation, “Tomahawk,” or “Pickaxe” method, can be employed for patients trapped in a vehicle [49] or those with spondylarthrosis [50]. However, this method requires good experience and should only be used after proper training, such as on a manikin [51]. Video laryngoscopes and video intubating laryngeal masks can successfully facilitate face-to-face endotracheal intubations [49,52,53,54]. There are also publications describing the possibility of using the Macintosh laryngoscope in face-to-face intubations [55]. Nonetheless, in our opinion, video laryngoscopes are better suited for this technique, achieving higher effectiveness, a shorter intubation time, and convenience for the anesthetist. The inverse intubation can be performed by one person successfully and does not require an assistant (Figure 9) [55]. The anesthetist, standing on the left side of the patient, can hold the video laryngoscope with their right hand and insert the tube with the left one. Alternatively, they can introduce the device with the left hand (as in the traditional approach) and relocate the video laryngoscope to the right hand after obtaining a satisfactory visualization of the larynx, then insert the intubation tube with the left hand. Channeled video laryngoscopes can also be used for this method, producing good results [56]. Regardless of the method, a video laryngoscope is undoubtedly a better choice than the traditional Macintosh device.

#### 3.2.5. Endotracheal Intubation through Supraglottic Devices Using Intubation Aids

Supraglottic devices can serve as a conduit for endotracheal intubation by blindly inserting an endotracheal tube or by using fiberoptic scopes as aids. There are specially designed SADs for blind intubation, such as LMA FastTrach, which has a high success rate of 92–99% [57]. However, the success rate of blind intubation through supraglottic devices varies depending on the type of device used, ranging from 20% for LMA Classic to 82% for Ambu Aura-I and 24–78% for Air-Q and iLT [58,59,60]. To increase the success rate of blind intubation through supraglottic devices, airway aids, such as bougies or Aintree intubation catheters, can be used [61].

Studies conducted on patients (not manikins) using the Aintree catheter as a guide for the endotracheal tube while intubating through a supraglottic device have shown an increased success rate of 70–90%. However, when using only a bougie without a bronchoscope, the success rate achieved ranged from 0–82%. Some authors suggest that blind intubation using a bougie as a conduit for intubation through a supraglottic device should not be recommended [61,62].

In our study using a manikin model, we observed similar results regarding the success of intubation without a bougie. The success rates for efficient intubation without a bougie were as follows: LMA-30%, Igel—67%, AmbuCurve LMA—20%, Air-Q—23%, and Cobra PLA—30%. The use of a tracheal tube introducer did not improve the effectiveness of intubation management with supraglottic airway devices; in fact, it diminished it. The success rates of blind intubation attempts using a bougie as an intubation conduit were low, with the Ambu LMA Curve at 3%, Cobra PLA at 10%, and iGEL at 27% [63]. There are some intubation aids, such as the Frova introducer, that may increase the success rate of blind intubation. However, there are limited studies on this topic [64]. Therefore, it is not recommended to use a bougie as conduit for blind intubation through SAD.

Blind intubation through supraglottic devices generally is not recommended for elective intubation for scheduled procedures. It may be attempted for emergency intubations when the equipment for visualizing the glottis, such as fiberoptic scopes, is not available, and securing the airway with endotracheal intubation is necessary. An alternative to fiberoptic intubation through supraglottic devices may be the use of video intubation stylets with supraglottic devices.

One example of such a multimodal airway approach is the use of the video intubation stylet Provu (Figure 10). The video intubation stylet Provu was designed to be used as a visualization aid for intubation using a standard laryngoscope. An additional advantageous feature of this tool is its movable tip, which may be shaped up and down during intubation efforts. This video intubation stylet may be used together with a supraglottic device for intubation through its lumen (Figure 11). Because of visualization and the possibility of directing the tip of the device into the glottis, the success rate of intubation can be increased similarly to using a fiberoptic scope in this technique. We used the video intubation stylet Provu for this purpose with good results.

## 4. Innovative Technologies

### 4.1. Artificial Intelligence in Airway Management

Anticipation and planning are fundamental principles in airway management. Current tests have limited and inconsistent diagnostic value [65], as the vast majority solely focus on predicting a difficult direct laryngoscopy, and all of them have low sensitivity and low negative predictive values [66], making none of them suitable for detecting an unexpected difficult airway.

Artificial intelligence (AI) is based on simulating and replicating the learning, reasoning, perception, and decision-making capacity of humans. AI systems use algorithms and mathematical models to process large amounts of data, learn patterns, adapt to new situations, and perform complex tasks. Artificial intelligence in medicine aims to improve the efficiency of health services, optimize resources, personalize treatment, and improve the quality of medical care in general. Some of the main artificial intelligence technologies used in the field of health are (1) machine learning: this involves training algorithms to learn patterns and make predictions or decisions. Algorithm learning improves with experience, allowing it to achieve its goal of process optimization. (2) Deep learning: a subset of machine learning that uses artificial neural networks with multiple layers to process complex data and extract high-level features. It is often used for tasks, such as image recognition. The objective of this algorithm is to sort, classify, and identify anomalies in patterns. One notable benefit of deep learning is its capacity to promptly detect data patterns that are typically overlooked by human experts [67]. (3) Neural networks: these are computational models inspired by the human brain’s structure and functioning. In the presence of errors, neural networks adjust the process and repeat it until the error decreases.

These systems can be trained to recognize images and make decisions based on the data. Thus, different trials have focused on their use in the field of airway examination to predict difficult airways [67,68,69,70,71]. Machine learning can predict adult difficult airways [72].

Cuendet et al. [71] presented the first completely automatic and noninvasive face-analysis system for detecting difficult intubations. It achieved an area under the curve (AUC) of 81.0% in a simplified experimental setup and 77.9% in a clinical setting.

Hayasaka et al. [69] developed an AI model based on a convolutional neural network to classify tracheal intubation difficulties for inexperienced medical staff based on the facial images of patients. This model establishes a connection between the facial image and the actual level of intubation difficulty. The results of the study showed an accuracy of 80.5%, sensitivity of 81.8%, specificity of 83.3%, and AUC of 0.864 (95% CI: 0.731–0.969). Likewise, Aguilar et al. [70] proposed a mobile application that employed a convolutional neural network to detect a difficult airway by classifying patients into two categories based on their Mallampati scores: 1–2 (low risk of a difficult airway) and 3–4 (higher risk). The app achieved an accuracy of 88.5% and a sensitivity and specificity of 90% each.

Tavolara et al. [68] conducted a study on the identification of difficult tracheal intubation cases using deep learning techniques applied to frontal face images. Their developed model achieved an AUC of 0.7105 when evaluated using a cohort consisting of 76 difficult-to-intubate patients and 76 easy-to-intubate patients. In comparison, two conventional bedside tests yielded AUC values of 0.6042 and 0.4661, respectively.

Wang et al. [67] used a semi-supervised deep-learning method to train and test an AI model for difficult airway prediction using images of 1000 patients captured from 9 specific viewpoints. The accuracy, sensitivity, specificity, and AUC were found to be 90.00%, 89.58%, 90.13%, and 0.9435, respectively.

Artificial intelligence shows promise for airway assessments; however, further evidence is needed, particularly regarding the cost-effectiveness of the technology.

### 4.2. Three-Dimensional Technology

The use of three-dimensional visualization, modeling, and printing techniques has revolutionized healthcare by providing detailed representations of anatomical structures.

Connor et al. [73] utilized 3D face-maker software to create a three-dimensional model of the head based on front and side face images from sixty-one patients. To identify patients with difficult intubations, they employed a feature selection model followed by logistic regression. The most effective model consisted of three facial parameters and thyromental distance. The computer model achieved a sensitivity of 90%, specificity of 85%, and an area under the curve (AUC) of 0.899.

Computed tomography (CT) and magnetic resonance imaging (MRI) produce detailed cross-sectional images that allow for the generation of 3D models and the visualization of anatomical structures in a comprehensive and realistic manner. This ability enables the improved assessment of complex airways, including those with tumors, congenital anomalies, or iatrogenic distortions, resulting in enhanced treatment planning and potential improved outcomes [74].

Furthermore, 3D technology has found applications in the fields of virtual reality and augmented reality allowing the interactive practice and training for airway management in a simulated environment [75,76].

On the other hand, 3D-printing technology allows for the creation of patient-specific anatomical models using the data acquired from CT scans to generate three-dimensional airway models using 3D rendering software, such as Pixmeo or OsiriX. These models can be development for simulation and planning purposes in the management of complex airways [74,77,78].

This innovative technology still has great potential to be integrated into the field of airway management.

The incorporation of various types of airway devices and their varieties has enriched the airway armamentarium. The multimodal approach, consisting of a combination of different types of devices, has further diversified the possibility of achieving success in airway management. Non-standard methods and devices for airway management are not a standard procedure in daily practice. However, they may be used in special circumstances when standard techniques fail or are not available. From the authors’ perspectives, there are several new technologies in airway management that may be introduced more widely in the future, such as the use of ultrasound in, robotic intubation, and artificial intelligence [4]. Although they are not yet new technologies and non-standard approaches, we assume that, in the near future, they may become a standard practice.

The advantages of modern methods of airway management compared to the traditional approaches are primarily associated with improved safety for patients. For instance, the integration of video laryngoscopes into daily medical practice can lead to a reduction in the complications related to issues, such as unrecognized esophageal intubations [79], as well as decreased trauma associated with intubation attempts [80]. When compared to standard laryngoscopes, video laryngoscopes exhibit a higher rate of first-pass success [80,81], a crucial factor, particularly in challenging patient cases, such as trauma patients or individuals who rapidly desaturate, such as those with obesity [82], obstetrics patients [83], and neonates [84]. However, the widespread adoption of video laryngoscopes in daily practice might provide limitations, notably in terms of costs. Some countries still find it expensive to equip every ambulance with video laryngoscopes. A potential solution to this problem involves the utilization of custom-made video laryngoscopes, as discussed in Section 3.2.2, or the exploration of alternative devices that can enhance the success rate of endotracheal intubations when video laryngoscopy is not available. One such device worth mentioning is the VieScope laryngoscope (Adroit Surgical, Oklahoma City, OK, USA), as depicted in Figure 12. The VieScope represents a novel laryngoscope design, featuring a straight, shielded, and illuminated tube. It was specifically developed for tracheal intubation using a gum elastic bougie and employing the paraglossal technique. Several studies have affirmed the effectiveness of the VieScope laryngoscope in improving the success rate of endotracheal intubation [85,86,87,88,89,90,91].

## 5. Conclusions

In conclusion, we assumed that the introduction of nonstandard airway management techniques or devices may be helpful to overcome some problems when using standard methods. Those techniques or devices may not be suitable for daily practice but may be useful for some groups of patients or in some clinical situations. Therefore, the knowledge of these techniques or devices is, in our opinion, valuable.

Take-home messages:Non-standard methods and devices for airway management can enhance patient safety in specific cases; however, they cannot substitute the recommended methods.The incorporation of these non-standard techniques and methods into clinical practice might assist in addressing the drawbacks or issues associated with standard methods in particular clinical scenarios.Innovative technologies in airway management should undergo further evaluations, as they can potentially become integral to standard airway management protocols in the future.

## Figures and Tables

**Figure 1 healthcare-11-02468-f001:**
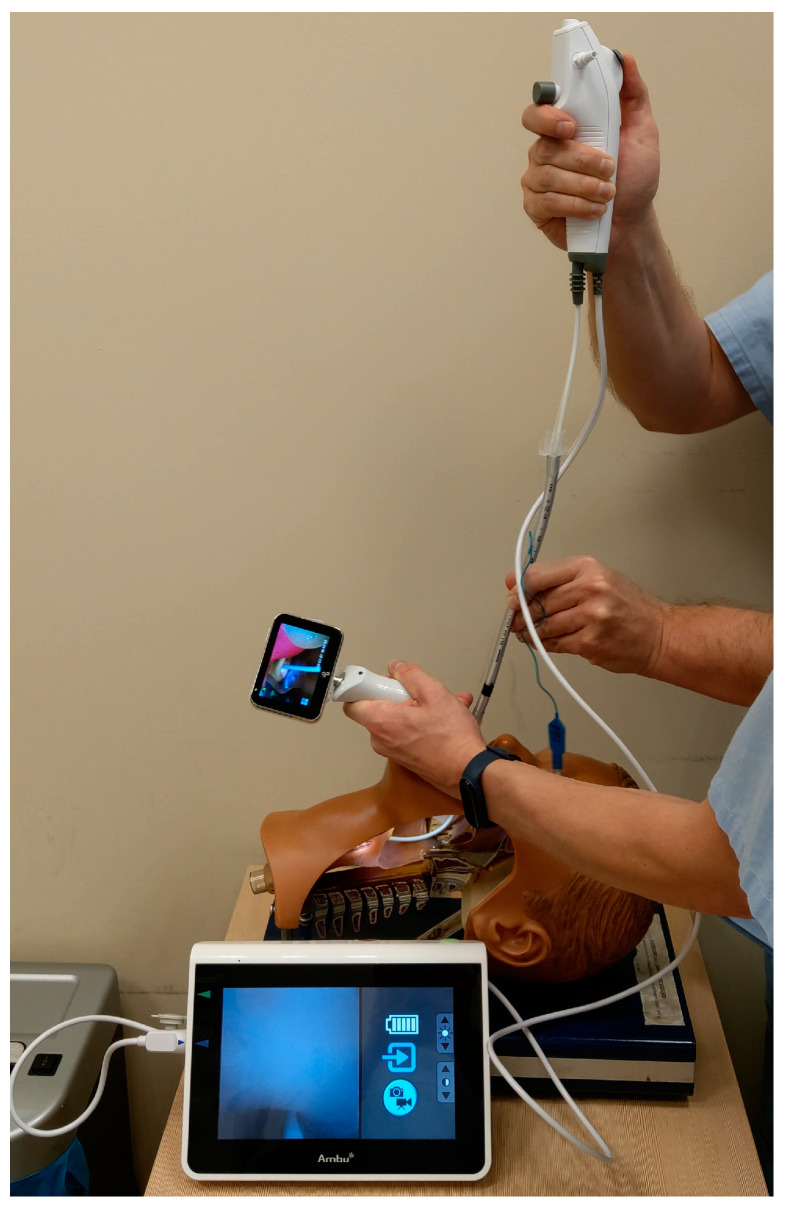
Combined use of bronchoscope and videolaryngoscope for endotracheal intubation (authors own materials).

**Figure 2 healthcare-11-02468-f002:**
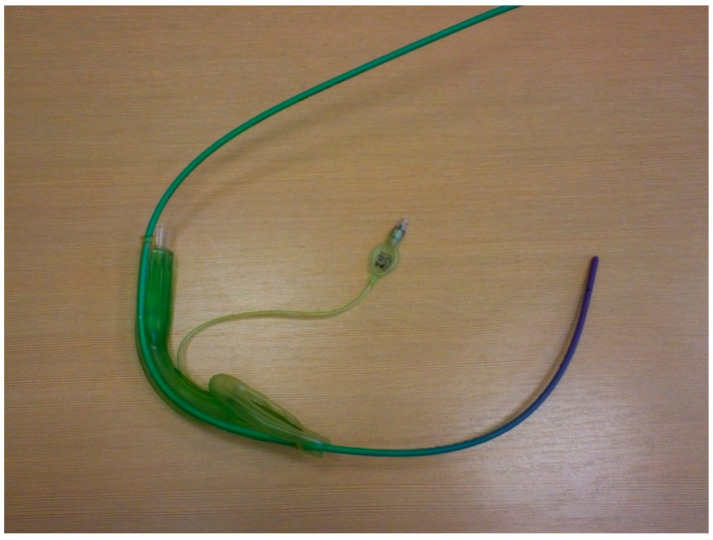
Bougie-assisted laryngeal mask insertion. Ambu AuraGain (Ambu A/S, Ballerup, Denmark) and standard bougie intubation wire (Sumi, Sulejowek, Poland) (source—authors own materials).

**Figure 3 healthcare-11-02468-f003:**
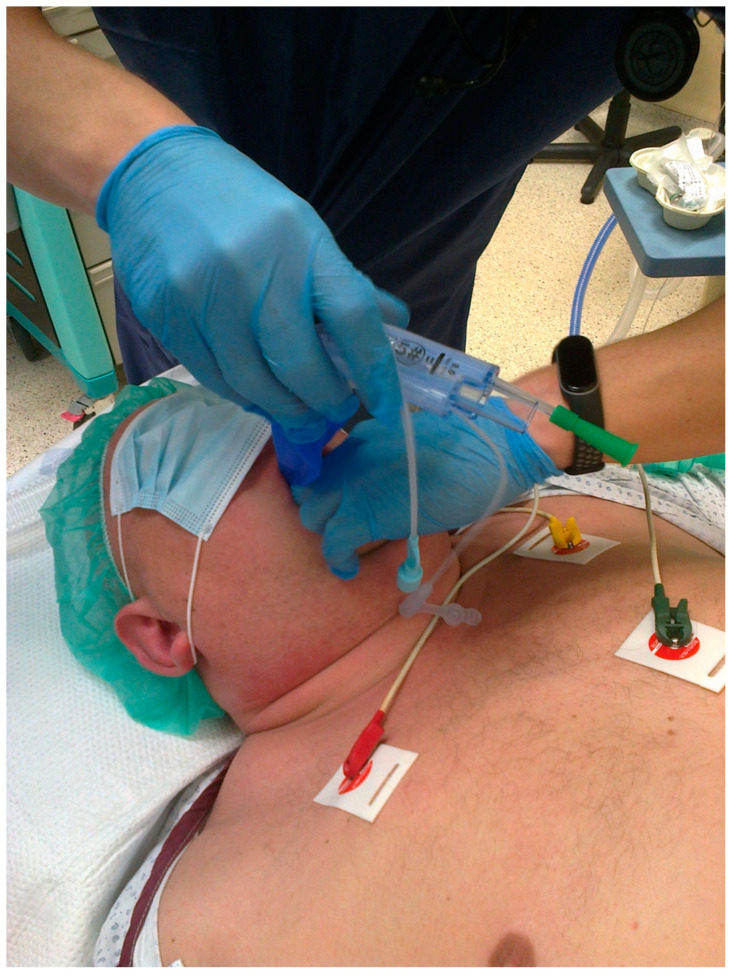
Gastric catheter laryngeal mask-assisted insertion. Ambu AuraGain (Ambu A/S, Ballerup, Denmark) and standard gastric catheter (Sumi, Sulejowek, Poland). Source—authors own materials.

**Figure 4 healthcare-11-02468-f004:**
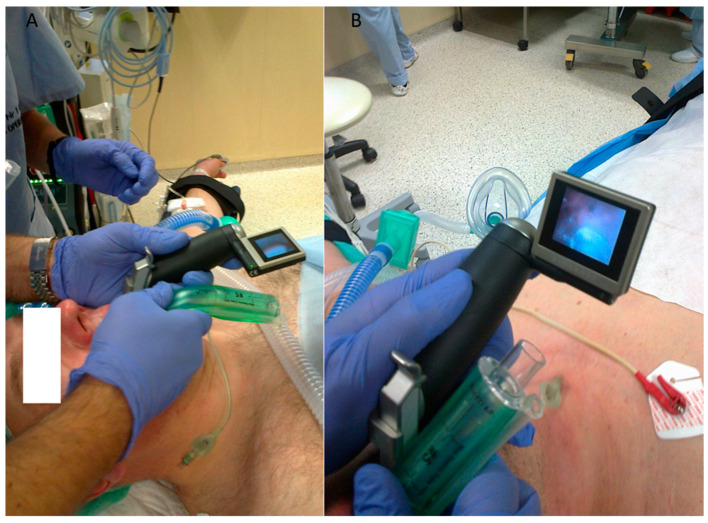
Videolaryngoscope-assisted laryngeal mask insertion: (**A**) insertion of laryngeal mask Ambu AuraGain (Ambu A/S, Ballerup, Denmark) using McGrath series 5 (AircraftMedial, Edinburgh, UK), (**B**) view of cuff of laryngeal mask on videolaryngoscope monitor. Source—authors own materials.

**Figure 5 healthcare-11-02468-f005:**
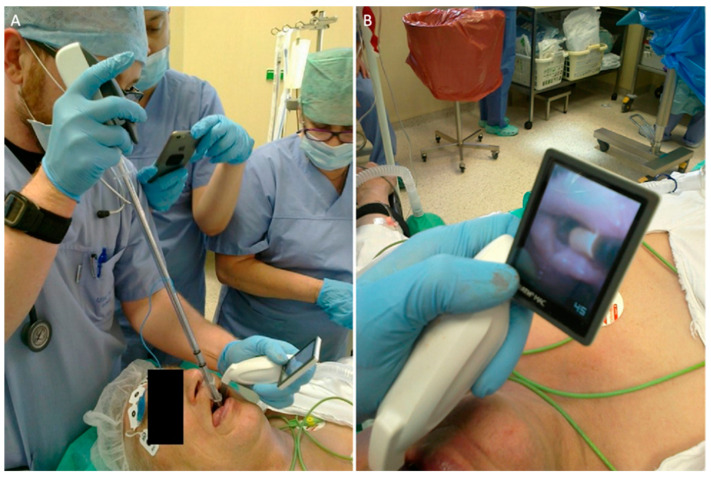
Combined use of the McGrath Mac video laryngoscope and the Goldfinger as a dynamic stylet: (**A**) insertion of Goldfinger stylet under control of videolaryngoscope, (**B**) view of tip of Golfinger in glottis obtained on monitor of videolaryngoscope. (Source—authors own materials).

**Figure 6 healthcare-11-02468-f006:**
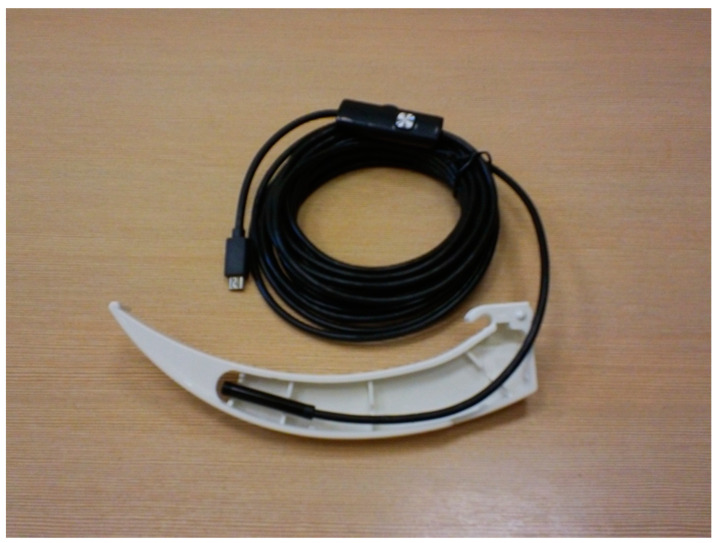
A custom-made device using a USB waterproof camera for smartphones attached to a standard single-use laryngoscope blade (Sinmed, Przyszowice, Poland). (Source—authors own materials).

**Figure 7 healthcare-11-02468-f007:**
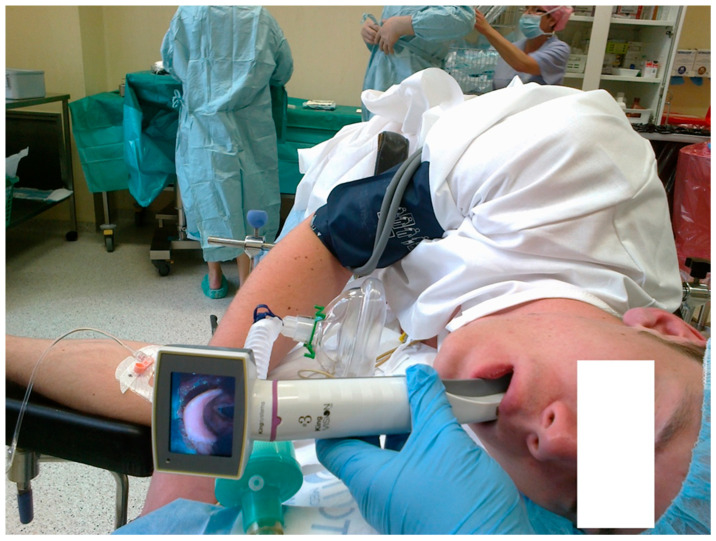
Tracheal intubation in the lateral position using a KingVision video laryngoscope (Ambu, Amsterdam, The Netherlands). Source—authors own materials.

**Figure 8 healthcare-11-02468-f008:**
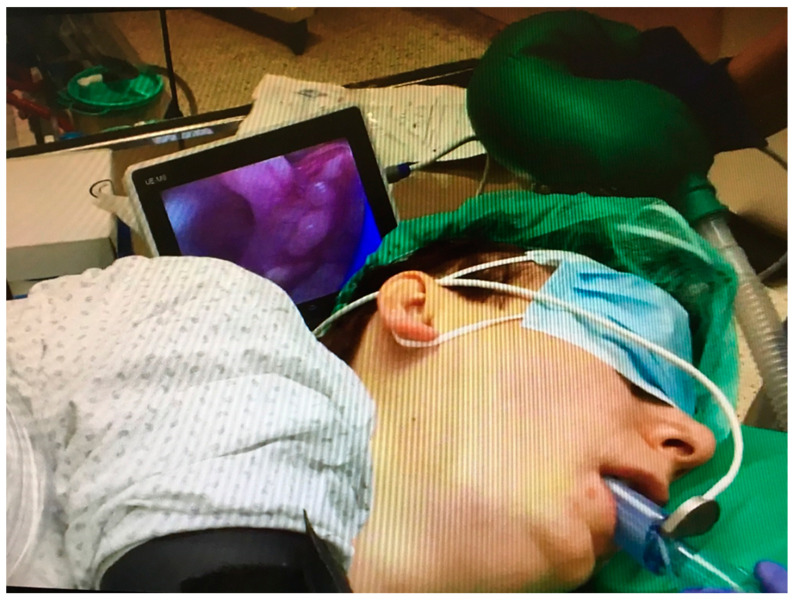
Tracheal intubation in the lateral position using the SaCo Video Laryngeal Mask (UE Medical Corporation, Zhejiang, China). Source—authors own materials.

**Figure 9 healthcare-11-02468-f009:**
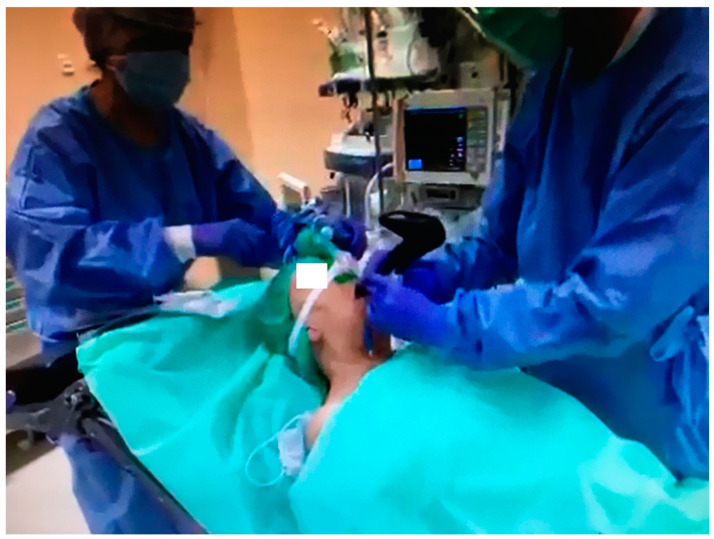
Face-to-face intubation. Source—authors own materials.

**Figure 10 healthcare-11-02468-f010:**
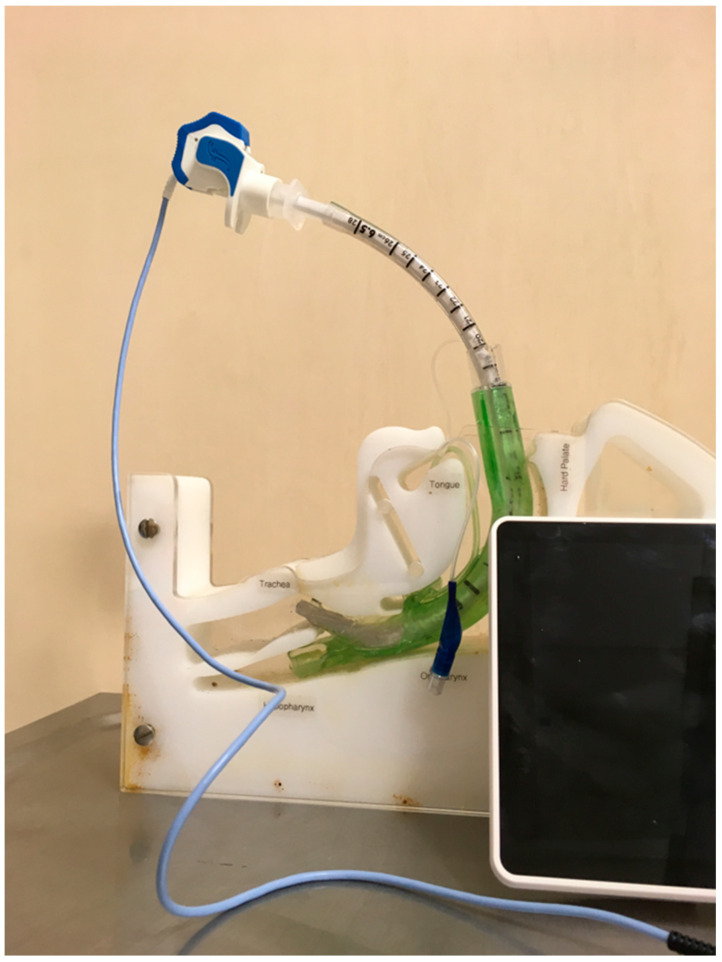
Inserted LMA with ProVu intubation stylet. (Source—authors own materials).

**Figure 11 healthcare-11-02468-f011:**
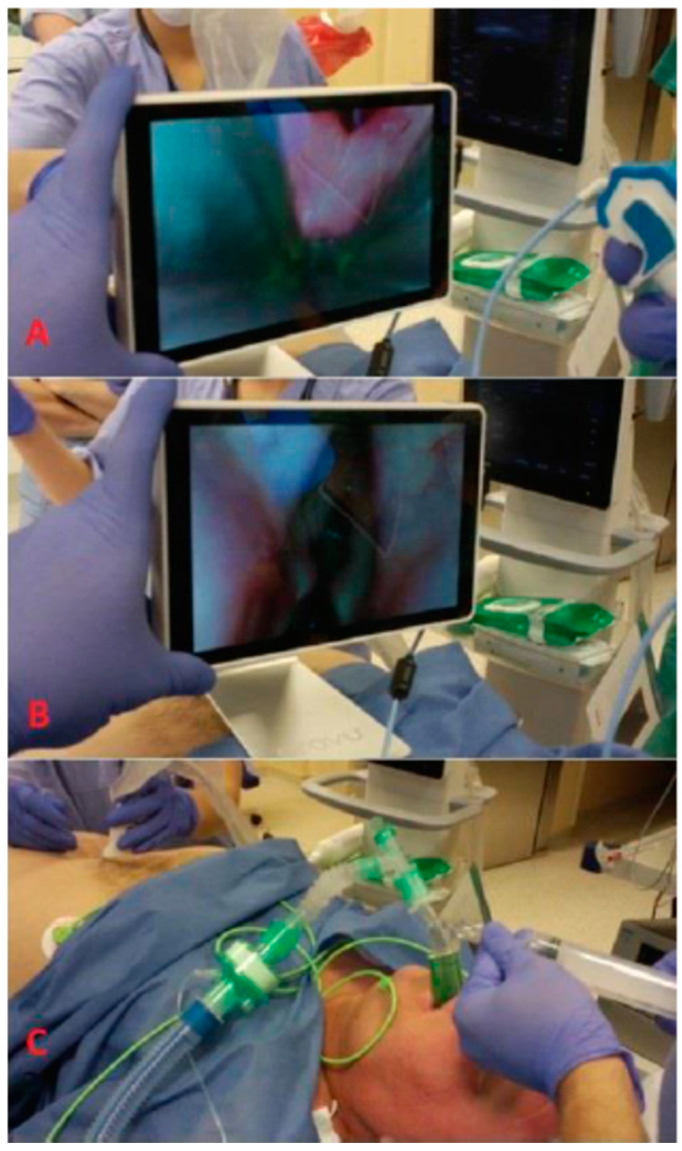
Intubation through SAD using ProVu intubation stylet: (**A**)—airways secured via LMA; (**B**)—view of the vocal cords; (**C**)—patient successfully intubated through LMA (source—authors own materials).

**Figure 12 healthcare-11-02468-f012:**
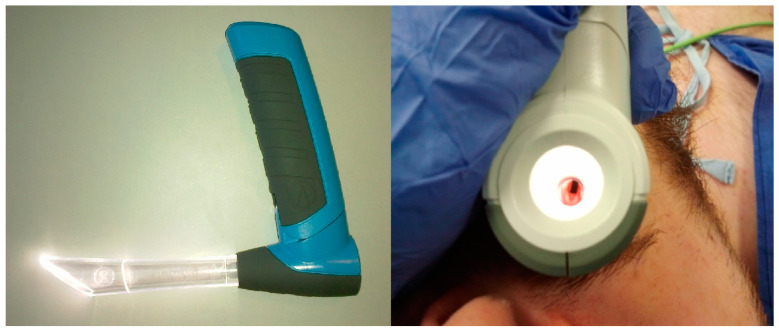
VieScope laryngoscope (Adroid Surgical, Oklahoma City, OK, USA). Source—authors own materials.

## Data Availability

Not applicable.

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
