# Peer review of "New Devices, Innovative Technologies, and Non-Standard Techniques for Airway Management: A Narrative Review"

_healthcare, 2023, doi:10.3390/healthcare11182468_

Round 1
Reviewer 1 Report
Thank you for the opportunity to review an article that features new techniques and devices that can help overcome the difficulties of difficult intubation.
The title of the manuscript reflects its subject. It is a review of reports on the introduction of videolaryngoscopes into practice to perform intubation in a patient requiring maintenance of airway patency. The article discusses the construction and possibilities offered by the new devices used for intubation and vividly demonstrates their use and effectiveness, especially when intubating stressed patients. The article is a synthesis of publications with a level of usefulness.
The review paper presents the use of new technologies that not only enable visualization of the intubation process, but also facilitate it and minimize the risk to the patient. Currently, they are moving towards inclusion to increase the progress of knowledge of the recommended airway management in "difficult" members entering the life of intubation to secure the airway. The paper collects thematic data from new intubation devices and techniques. - My introduction of the addition of figures to the paper
Deveral comments and suggestions for the authors:
-
The results of the literature review should be included in the summary of the article, preferably in the form of a table,
-
No author of origin figures in the article,
-
From the point of view of the topic and purpose of the work, it would be worthwhile to supplement with the benefits for the patient of the modern methods used in relation to the traditional ones.
-
Add arguments for introducing videolaryngoscopy into daily practice as well as point out limitations.
Author Response
The results of the literature review should be included in the summary of the article, preferably in the form of a table
-
- Thank you very much for this comment, however, this paper is not typical literature review, so in our opinion it may be difficult to summarize such wide range of citation on several completely diffirent topics, therefore we decided not to perform summarize of literature review. Citations are to confirm our observations, not to compare them with eachother.
No author of origin figures in the article
- all figures are our own materials,
From the point of view of the topic and purpose of the work, it would be worthwhile to supplement with the benefits for the patient of the modern methods used in relation to the traditional ones.
-
- thank you for this valuable remark, we added paragraph dealing with this topic
Add arguments for introducing videolaryngoscopy into daily practice as well as point out limitations -
-
- thank you for this valuable remark, we added paragraph dealing with this topic.
Reviewer 2 Report
Thank you for the opportunity to revise the manuscript by Gaszyński et al. on new devices, innovative technologies, and non-standard techniques for airway management. This review is very well written and extremely interesting and novel. I have only minor issues to be addressed:
- I believe paragraphs should be organized differently in order to highlight the differences between supraglottic devices placement and endotracheal intubation. Moreover, I would suggest to place the paragraph on innovative technologies, including the use of artificial intelligence, at the end of the paper. Finally, the title of paragraph 2 should be in capital letters.
- Line 253. Besides the use of stylets or bougies, the use of fibro-bronchoscopy has been also tested as a possible strategy to overcome the issue of difficult passage of the endotracheal tube through the vocal cords during laryngoscopy (DOI: 10.22514/sv.2023.053 - DOI: 10.22514/sv.2023.073). Please briefly discuss and add these 2 references.
- Please provide a conclusion section with the take home messages of your review.
Author Response
Thank you for the opportunity to revise the manuscript by Gaszyński et al. on new devices, innovative technologies, and non-standard techniques for airway management. This review is very well written and extremely interesting and novel. I have only minor issues to be addressed:
- I believe paragraphs should be organized differently in order to highlight the differences between supraglottic devices placement and endotracheal intubation. Moreover, I would suggest to place the paragraph on innovative technologies, including the use of artificial intelligence, at the end of the paper. Finally, the title of paragraph 2 should be in capital letters.
- - we moved paragraph on innovative technologies to the end of manuscript and corrected other issues.
- Line 253. Besides the use of stylets or bougies, the use of fibro-bronchoscopy has been also tested as a possible strategy to overcome the issue of difficult passage of the endotracheal tube through the vocal cords during laryngoscopy (DOI: 10.22514/sv.2023.053 - DOI: 10.22514/sv.2023.073). Please briefly discuss and add these 2 references.
- - we added the references and paragraph about this method
- Please provide a conclusion section with the take home messages of your review
- - we added take home methods.
Reviewer 3 Report
General comment:
I read the manuscript with the interest because it is my area of clinical expertise, and the title calls for attention.
It is a narrative review of non-standard new devices and techniques for airway management.
The review is rather clear, comprehensive and of relevance of the filed of airway management. A gap in clinical use of the described devices, innovative technologies and non- standard techniques for airway management is well identified, however the authors should make more emphasis on a gap in knowledge and identify relevant areas how to improve underusage of these techniques in clinical practice.
Although there are similar published reviews recently, this current review is still relevant and of interest to help update busy anaesthesiologists. However, it should point out more scientific evidence for being of the interest to scientific community.
The cited references are mostly recent publications, although there are some of later date ( Ref 35, Ref 70) that should be updated or omitted. In addition, the authors describe their clinical experience in the manuscript without the note of publishing or not publishing this experience before, that is not appropriate. There are some self- citations but in acceptable number because it justifies the expertise of the authors.
The statements and conclusions drawn are rather coherent ad supported by the listed citations, but should be improved according to the specific comments. The authors describe mostly the advantages of new devices/techniques without evidence about their efficacy. The authors should identify clearly if evidence for a specific device/techniques is missing, give their own opinion for a specific described device/techniques and try to avoid generalisations.
There are only figures that are appropriate for the content of the manuscript but should be improved by adding ownership or their source. If being own photos with patients, it should be at least institutional’s approval if there is no patient informed consent. Additionally legends to figures should not include abbreviations or these should be explained, otherwise not easy to understand.
Specific comments:
Title
Line 2-3. Title can be improved by adding the type of review; is it possible to change the phrase “ New devices” into more specific for this review
Abstract
Line 11-16 Abstract- should be more specific for the current review; when read in isolation it is not clear on which devices and techniques the review refers.
Line 15-16 “… , but can be highly effective in overcoming a difficulty…” this conclusion is misleading and not clear because, as mentioned before , the authors do not mention any specific new device in the abstract. In addition, this conclusion is not adequately supported by citations in the main manuscript.
Keywords
Teaching is irrelevant keyword.
The manuscript does not refer to education and teaching. It refers to clinical practice.
Introduction
Line 41 The authors should be more specific in explaining the aim and “ some new devices and techniques”
Line 42-43 Citation missing for supporting the statement “ that can be highly effective in overcoming a difficult related to airway management”
There is no part on methodology of writing the manuscript ( e.g. search of the literature) and no explanation how and why the authors chose to describe the specific devices , technologies and techniques .
Line 191-192. Not clear and scientific evidence missing “ Authors used gastric catheter assisted insertion of laryngeal mask with good results”.
Line 203- 210 The whole paragraph 4.1.2. should be reevaluated for relevance of being new techniques. Ref 35 is from 1995. Is there any other evidence of its clinical usage except this one citation? Why not?
Line 208 Statements in brackets not clear
Line 226-Line 230 Citation missing or clear explanation how the authors evaluated the use of different video laryngoscopes, which VLS, which complications..
Line 280- Line 290 is the case report. Not clear was it published or unpublished. Not really clear why the authors describe their case report within this narrative review if not published
Line 305 not clear on which cases and clinical practice the authors refer?
Line 320-321 Citation missing
Line 339 Not clear “ many “ . In Reference 57 there is no figure how many patients.
Line 362-363 it is not clear if this statement misses citation or the opinion of the authors
Line 382. 70-90%- Citation missing
Line 383 0-82% - citation missing
Paragraph 4.2.3. refers to blind intubation. Is it really of relevance for this review?
Line 385 Statement and Ref 70 – dated 2012, this was narrative review based on literature 1948-July 2011, is there any update?
Line 407-408 citation or clarification needed ; the statement “ We used…with good results”, is not appropriate for scientific text
Conclusion paragraph should be rewritten and the conclusion drawn only on the devices and techniques described in the main body.
Line 417-419 The statement “Non-standard methods and devices….should not be used in clinical practice” is not clear in the context of the current review. The next statement is not clear because it does not specify in which special circumstances it would be allowed. Is it the authors’ opinion or evidence based.
Line 421 “ ultrasound” and “robotic intubation” are not described in the main body of the manuscript
Line 422-425 not belong to conclusion
Line 426-428 Final statement not drawn from the main body. If want to conclude with POCUS, ultrasound technology should be described before in the main body with the evidence for supporting its change from being non-standard to becoming standard practice.
Minor editing of English language required
Author Response
The authors should make more emphasis on a gap in knowledge and identify relevant areas how to improve underusage of these techniques in clinical practice.
However, it should point out more scientific evidence for being of the interest to scientific community.
The cited references are mostly recent publications, although there are some of later date (Ref 35, Ref 70) that should be updated or omitted. In addition, the authors describe their clinical experience in the manuscript without the note of publishing or not publishing this experience before, that is not appropriate. There are some self- citations but in acceptable number because it justifies the expertise of the authors.
The statements and conclusions drawn are rather coherent ad supported by the listed citations, but should be improved according to the specific comments. The authors describe mostly the advantages of new devices/techniques without evidence about their efficacy. The authors should identify clearly if evidence for a specific device/techniques is missing, give their own opinion for a specific described device/techniques and try to avoid generalisations.
There are only figures that are appropriate for the content of the manuscript but should be improved by adding ownership or their source. If being own photos with patients, it should be at least institutional’s approval if there is no patient informed consent. Additionally, legends to figures should not include abbreviations or these should be explained, otherwise not easy to understand.
Most of the techniques presented in this manuscript still have emerging evidence. Therefore, even though our initial goal was to conduct a rapid systematic review, we proceeded with a narrative review. We have supplemented the gap in scientific evidence with our practical clinical experience related to these methods and devices. We have made this clear to the reader (L56-L57).
Specific comments:
Title
Line 2-3. Title can be improved by adding the type of review; is it possible to change the phrase “New devices” into more specific for this review.
Done. We believe that if the title becomes more explicit about the discussed devices, it would significantly increase its length. This could potentially reduce its appeal.
Abstract
Line 11-16 Abstract- should be more specific for the current review; when read in isolation it is not clear on which devices and techniques the review refers.
Done.
Line 15-16 “…, but can be highly effective in overcoming a difficulty…” this conclusion is misleading and not clear because, as mentioned before, the authors do not mention any specific new device in the abstract. In addition, this conclusion is not adequately supported by citations in the main manuscript.
We have replaced the word "can" with "could".
Keywords
Teaching is irrelevant keyword. The manuscript does not refer to education and teaching. It refers to clinical practice.
We have removed this keyword.
Introduction
Line 41 The authors should be more specific in explaining the aim and “some new devices and techniques”
Done.
Line 42-43 Citation missing for supporting the statement “that can be highly effective in overcoming a difficult related to airway management”
We have replaced the word "can" with "could".
There is no part on methodology of writing the manuscript (e.g. search of the literature) and no explanation how and why the authors chose to describe the specific devices, technologies and techniques .
Done.
Line 191-192. Not clear and scientific evidence missing “Authors used gastric catheter assisted insertion of laryngeal mask with good results”.
We have removed “with good results”
Line 203- 210 The whole paragraph 4.1.2. should be revaluated for relevance of being new techniques. Ref 35 is from 1995. Is there any other evidence of its clinical usage except this one citation? Why not?
this citation is the only one on this method, and we gave it as an example to introduce further disscussion on other methods
Line 208 Statements in brackets not clear
We have removed the content Statements in brackets
Line 226-Line 230 Citation missing or clear explanation how the authors evaluated the use of different video laryngoscopes, which VLS, which complications.
- we removed this sentence because the data of our observation on this are not published yet
Line 280- Line 290 is the case report. Not clear was it published or unpublished. Not really clear why the authors describe their case report within this narrative review if not published
- removed
Line 320-321 Citation missing - they are unpublished data - added to text
Line 339 Not clear “ many “ . In Reference 57 there is no figure how many patients. - thank you for this remark. I do intubations with VL in lateral position as routine and I can not remember the number of pts I intubated this way, but it was now several hundert
Line 362-363 it is not clear if this statement misses citation or the opinion of the authors - added citation
Line 382. 70-90%- Citation missing - added
Line 383 0-82% - citation missing- added
Paragraph 4.2.3. refers to blind intubation. Is it really of relevance for this review? - we belive it is som kind of introduction and explanation to present other deices like ProVu
Line 385 Statement and Ref 70 – dated 2012, this was narrative review based on literature 1948-July 2011, is there any update?
We have added a newer reference.
Line 407-408 citation or clarification needed; the statement “ We used…with good results”, is not appropriate for scientific text
We have removed “with good results”.
Conclusion paragraph should be rewritten and the conclusion drawn only on the devices and techniques described in the main body.
Done.
Line 417-419 The statement “Non-standard methods and devices…. should not be used in clinical practice” is not clear in the context of the current review. The next statement is not clear because it does not specify in which special circumstances it would be allowed. Is it the authors’ opinion or evidence based.
We have removed “and should not be used in daily practice”.
Line 421 “ultrasound” and “robotic intubation” are not described in the main body of the manuscript
That's correct. Our goal wasn't to describe them in-depth within the text. We simply wanted to present them as relatively recent techniques that will gain prominence over time. We consider it appropriate not to omit these two technologies.
Line 422-425 not belong to conclusion
We have included the conclusion section in a separate paragraph.
Line 426-428 Final statement not drawn from the main body. If want to conclude with POCUS, ultrasound technology should be described before in the main body with the evidence for supporting its change from being non-standard to becoming standard practice.
We have included the conclusion section in a separate paragraph.
Round 2
Reviewer 3 Report
The authors sent the new version in which they have incorporated most of my suggestions or have given adequate explanation.
My additional comment on this new version:
- There is duplication of sentences in the Abstract ( L14-19): the later sentence is sufficient and enough
“ The objective of this article is to describe some new devices and techniques that are not applied in daily practice, but that could be highly effective in overcoming a difficulty related to airway management. The aim of this narrative review is to describe some new devices as video laryngeal masks, articulated stylets, and non-standard techniques for laryngeal mask insertion and endotracheal intubation. Artificial Intelligence and 3 D Technology in airway management are also discussed.”
Minor editing of English language required
Author Response
Dear Reviewer,
thank you for comments.
We adjusted manuscript following your suggestions,